# Comparison of Pretreatment Methods for Salinity Gradient Power Generation Using Reverse Electrodialysis (RED) Systems

**DOI:** 10.3390/membranes12040372

**Published:** 2022-03-29

**Authors:** Jaehyun Ju, Yongjun Choi, Sangho Lee, Chan-gyu Park, Taemun Hwang, Namjo Jung

**Affiliations:** 1Environmental Technology Division Water Environment Center, Korea Testing Laboratory, 87, Digital-ro 26-gil, Guro-gu, Seoul 08389, Korea; jaehyunju@ktl.re.kr (J.J.); pcg6189@naver.com (C.-g.P.); 2School of Civil and Environmental Engineering, Kookmin University, Seoul 02707, Korea; choiyj1041@gmail.com; 3Korea Institute of Civil Engineering and Building Technology, 283, Goyangdae-ro, Ilsanseo-gu, Goyang-si 10223, Korea; taemun@kict.re.kr; 4Korea Institute of Energy Research, 200, Haemajihaean-ro, Gujwa-eup, Jeju-si 63357, Korea; njjeong@kier.re.kr

**Keywords:** reverse electrodialysis (RED), pretreatment, membrane fouling, pressure drop, excitation–emission matrix (EEM), PARAFAC

## Abstract

With the increasing concern about climate change and the energy crisis, the use of reverse electrodialysis (RED) to utilize salinity gradient power (SGP) has drawn attention as one of the promising renewable energy sources. However, one of the critical issues in RED processes is membrane fouling and channel blockage, which lead to a decrease in the power density. Thus, this study aims to improve our understanding of SGP generation by using RED by investigating the effect of pretreatment on the RED performance. Experiments were conducted by using a laboratory-scale experimental setup for RED. The low-salinity and high-salinity feed solutions were brackish water reverse osmosis (BWRO) brine from a wastewater reclamation plant, and a NaCl solution simulating seawater desalination brine. Several pretreatments were applied to the RED process, such as cartridge filter (CF), microfiltration (MF), ultrafiltration (UF), nanofiltration (NF), activated filter media (AFM), and granular activated carbon (GAC). The results indicate that the open-circuit voltage (OCV) and the power density were similar, except for in the NF pretreatment, which removed the dissolved ions to increase the net SGP. However, the pressure in the RED stack was significantly affected by the pretreatment types. The excitation–emission matrix (EEM) fluorescence spectroscopy and the parallel factor analysis (PARAFAC) quantified the organic compounds that are related to the stack pressure. These results suggest that the removal of both colloidal and organic matters by pretreatments is crucial for improving the RED performance by reducing the pressure that is increased in the RED stack.

## 1. Introduction

There is an increasing urge to explore sustainable energy from renewable sources because of the limited availability of fossil fuel resources and because of climate change caused by greenhouse gas emissions [1,2]. This has led to the development of renewable energy technologies that use solar energy, wind power, tidal energy, and geothermal energy [3]. An emerging energy source that has recently attracted attention is salinity gradient power (SGP) [3,4,5,6]. Aqueous solutions with different ion concentrations have different chemical potentials, and, thus, the mixing of such solutions results in SGP [4]. The theoretical mixing energy for 1 m^3^ of river water, and for 1 m^3^ of seawater, corresponds to 0.70 ~ 0.75 kWh [7]. Considering the quantities of fresh water and saline water that are mixed in estuaries, the potential amount of SGP in the world is estimated to be 1.9 TW [8].

SGP, itself, cannot be directly used, and it thus requires special techniques. One of them is pressure retarded osmosis (PRO), and another is reverse electrodialysis (RED) [5]. The role of PRO is to transform the osmotic pressure to mechanical power [3,9,10]. To obtain electricity, the PRO needs to have additional devices, such as power generators [11]. Otherwise, PRO ensures that the osmotic power is directly used in reverse osmosis (RO) desalination plants through energy recovery devices [12]. On the other hand, the role of RED is to convert the chemical potential difference to electrical power. [13,14,15]. A stack of cation exchange membranes (CEMs) and anion exchange membranes (AEMs) are used in the RED system [8,16]. Depending on the applications, either PRO or RED may be used because of their strengths and weaknesses [3]. While PRO is preferentially considered to be combined with desalination plants [12], RED is considered a renewable energy source because of its advantage as a stand-alone system.

Unlike pressure-driven membrane systems, RED uses CEMs and AEMs, which have different properties from conventional polymeric membranes [17,18,19,20,21]. A RED module includes CEMs, AEMs, and spacers [14,22,23]. To generate a potential, high- and low-salinity solutions should flow through the channel between the CEMs and AEMs [13,14,24]. There are several factors that affect the performance of RED systems, including the properties of CEMs and AEMs [25,26], the module design, and the operating conditions [27,28].

To improve the performance of RED systems, a lot of works have been devoted to the fabrication of CEMs and AEMs, the spacer design, process optimization, and the configuration of hybrid processes [15,29]. Moreover, there have been investigations with regard to the preparation of electrodes and electrode solutions, as well as to the development of theoretical models [14,26,28,30,31]. Nevertheless, relatively few works have been performed to scale up and demonstrate RED systems. [28,32,33]. Because of a lack of pilot-scale data and experience, RED technology has not been widely accepted by the industry. More studies are required to fill the knowledge gap with regard to the design, operation, and maintenance of practical RED systems.

In this context, the purpose of this paper was to investigate the effect of pretreatment on the performance of a RED process. Reverse osmosis (RO) brine from a wastewater reclamation plant was used as the low-salinity feed solution. Since the contaminants in BWRO brine may result in membrane fouling [10,34], it is necessary to apply appropriate pretreatment methods to mitigate this [35]. Although there have been works on the fouling behaviors and mechanisms in RED membranes [36,37,38], little attention has been paid to optimizing the pretreatment methods in order to maintain the performance of the RED process.

Several pretreatments were applied to this feed solution, including cartridge filter (CF), microfiltration (MF), ultrafiltration (UF), nanofiltration (NF), granular activated carbon (GAC), and activated filter media (AFM). Both theoretical and experimental approaches were used to identify the factors that affect the power density and the stack pressure of the RED system. Fluorescence excitation–emission matrix (EEM) spectroscopy coupled with parallel factor analysis (PARAFAC) was conducted to quantitatively analyze the effect of pretreatment in terms of organic matter compositions. The novelty of this work lies in a systematic analysis of RED performances under various pretreatment conditions, which provide insight into the effective design and operation of RED processes.

## 2. Materials and Methods

### 2.1. Lab-Scale RED System

All the RED experiments were carried out in a laboratory-scale system, which is schematically illustrated in Figure 1a. The high-salinity solution and low-salinity solution were supplied to the RED stack by using microflow pumps (Labpinon, Seoul, Korea). There were two electrodes at the ends of both endplates, which measured the potential across the stack. An electrode rinse solution (ERS), and a mixture of 0.1 M K_3_[Fe(CN)_6_] and 0.1 M K_4_[Fe(CN)_6_] (Sigma Aldrich, St. Louis, MO, USA) was used. A source meter (Keithley 2401, SnM, Seoul, Korea) measured the open-circuit voltage (OCV) and the power. The electrical conductivities of the high-salinity and low-salinity solutions were measured by using conductivity meters (WTW 3420, Oberbayern, Germany). The photography of the lab-scale RED system is presented in Figure 1b. The operating conditions for the RED experiments are also presented in Table 1.

The CEMs and AEMs were supplied by Fujifilm (Type-1, Fujifilm Manufacturing Europe, Tilburg, The Netherlands). Table 2 presents the characteristics of these membranes. The thicknesses, area resistance, and transport number of the CEM were 125 μm, 1.87 Ω·cm^2^, and 0.952, respectively. Those of the AEM were 124 μm, 1.08 Ω·cm^2^, and 0.963, respectively. These CEMs and AEMs were placed in the RED stack.

As illustrated in Figure 2, the RED stack was assembled between the anode and the cathode. The CEMs and AEMs are stacked between the electrodes. The number of cell pairs was 10, and the effective membrane area was 0.0019 m^2^ per cell. The membranes were separated from each other by a gasket and a spacer. The spacer (DS Mesh, Seoul, Korea) has an open area of 81.3%, and a thickness of 100 μm [39]. Between the shielding membrane (CEM) and the electrode, a different spacer (thickness 0.5 mm, Sefar, Seoul, Korea) was used [40,41]. The endplate was made of acrylic plastic. The details on the RED stack used in this study are available in the literature [42].

### 2.2. Preparation of High-Salinity and Low-Salinity Solutions

A synthetic brine of seawater reverse osmosis (SWRO) was used as the high-salinity (HS) solution, which was prepared using sodium chloride (Samchun, Seoul, Korea) and deionized water. The NaCl concentration of the HS solution was 1.0 M. A real BWRO brine from a wastewater reclamation plant was used as the low-salinity (LS) solution. The feed water to the BWRO process was the effluent of a municipal wastewater treatment plant.

### 2.3. Pretreatments

Two groups of pretreatment methods were applied to the raw feed water, including membrane-based pretreatments and conventional pretreatments. The membrane-based pretreatment methods used here include cartridge filter (CF), microfiltration (MF), ultrafiltration (UF), and nanofiltration (NF), which were chosen to investigate the effect of the rejection capability. Activated filter media (AFM) and granular activated carbon (GAC) were adopted as the conventional pretreatment methods to represent the media filtration and the adoption, respectively. Table 3 shows the summary of the pretreatment methods. The CF and MF membranes have nominal pore sizes of 5 and 0.22 μm, respectively. The UF and NF membranes have molecular weight cutoffs (MWCOs) of 100 kDa and 200–300 Da, respectively. The operating pressures for the CF, MF, UF, and NF were 0.1, 0.5, 1, and 2.5 bar, respectively. A bench-scale experimental system for membrane filtration was utilized for these pretreatments, as depicted in Figure 3a. The AFM, which is an activated filter media that is prepared from recycled green glass bottles, was supplied by HOIMYUNG in Korea. Its particle size ranged from 0.4 to 1 mm, with an effective size of 0.46 mm. The GAC was provided by Sunghong-Lab in Korea, and its size ranges from 0.2 to 5 mm. The active surface areas and the iodine numbers of the AFM and the GAC are presented in Table 3. The hydraulic residence time of the GAC was adjusted to 12 min. As illustrated in Figure 3b, there were two tanks, one feed tank, two gear pumps, and one air pump in the experimental equipment for the AFM/GAC. Table 3 provides a summary of the pretreatment conditions.

### 2.4. Model Development

A mathematical model for RED systems was developed to interpret the experimental data. Three assumptions were made for this model:The current flow distribution is continuous;Only the difference in the ion concentrations between the HS and LS solutions is considered to calculate the open-circuit voltage (OCV) and the power density;The parameters used in the model are evaluated under average conditions between the inlet and the outlet [43].

On the basis of the Nernst equation, the open-cell voltage (*E_OCV_*) was evaluated: (1)$$E_{OCV}\left(i\right)=N_{m}\cdot \alpha_{CEM}\cdot \frac{R\cdot T}{z\cdot F}\ln\left(\frac{\gamma_{HC}\cdot c_{HC}}{\gamma_{LC}\cdot c_{LC}}\right)+N_{m}\cdot \alpha_{AEM}\cdot \frac{R\cdot T}{z\cdot F}\ln\left(\frac{\gamma_{HC}\cdot c_{HC}}{\gamma_{LC}\cdot c_{LC}}\right)$$
where $\alpha_{CEM}$ and $\alpha_{AEM}$ are the permselectivity of the CEM and AEM, respectively; *F* is Faraday’s constant (96,485 C·mol^−1^); *R* is the gas constant (8.31 J·mol·K^−1^); *T* is the temperature (K); *C* is the ion concentration (mol·m^−3^); *z* is the valence; and *γ* is the activity coefficient, which is determined by the Debye Hukel (0 < *C* < 1 M [44] or Pitzer (*C* > 1 M) equations [42]. The output voltage (*E_out_*) and the electrical current (*I*) are used to estimate the gross power (*P*):(2)$$P=E_{out}\cdot I$$

There is a relationship between the *E_out_* and *E_OCV_*: (3)$$E_{out}=E_{OCV}-R_{i}\left(x\right)\cdot I$$

The resistance of the RED stack (*R_i_*) is the sum of the resistances of all the cells in the stack. The internal losses in the cell pairs consist of ohmic and nonohmic resistances. Accordingly, *R_i_* can be experimentally examined by using the electronic load [45,46]: (4)$$R_{i}\left(x\right)=R_{ohmic}\left(x\right)+R_{non-ohmic}\left(x\right)$$

The information on the membrane characteristics and the compartment resistance of the HS and LS solutions allows the calculation of $R_{ohmic}$: (5)$$R_{ohmic}\left(x\right)=N_{m}\cdot \left(\frac{R_{CEM}}{1-\beta }+\frac{R_{AEM}}{1-\beta }+\frac{h_{HS}}{\varepsilon^{2}\cdot k_{HS}}+\frac{h_{LS}}{\varepsilon^{2}\cdot k_{LS}}\right)$$
where $R_{CEM}$ and $R_{AEM}$ are the membrane resistances (Ω·m^2^); β is the mask factor of the membrane; ε is the porosity of the spacers (-); $h_{HS}$ and $h_{LS}$ are the intermembrane distances; and $k_{HS}$ and $k_{LS}$ are the electric conductivities of the HS and LS solutions. $R_{non-ohmic}$ can be also examined by: (6)$$R_{non-ohmic}\left(x\right)=R_{\Delta c}\left(x\right)+R_{BL}\left(x\right)$$
where $R_{\Delta c}$ contributes to the resistance of the concentration change between the inlet and outlet. $R_{\Delta c}$ is estimated using the following equation:(7)$$R_{\Delta c}\left(x\right)=N_{m}\cdot \alpha \cdot \frac{R\cdot T}{z\cdot F\cdot j\cdot \frac{A}{m^{2}}}\ln\left(\frac{A_{LS}\left(x\right)}{A_{HS}\left(x\right)}\right)$$
where $A_{LS}\left(x\right)$ and $A_{HS}\left(x\right)$ is the area resistance due to the bulk concentration. $R_{BL}$ can be obtained by: (8)$$R_{BL}\left(x\right)=N_{m}\cdot \left(\frac{0.62t_{res}\cdot \frac{h_{sea}}{LL}}{sec}+0.05\right)$$
where $t_{res}$ is the resistance time inside the stack. $t_{res}$ is calculated by Equation (8).
(9)$$t_{res}=\frac{L\cdot b\cdot \delta \cdot \epsilon }{Q\left(x\right)}$$
where *b* is the width of the cell (m), and *L* is the length of the cell (m). A MATLAB code was developed by combining the above equations, which allowed for the calculation of the OCV and the power density.

### 2.5. EEM Analysis and PARAFAC Model

The EEM analysis was performed to characterize the organic matter in the raw water and the membrane. Using an EEM instrument (EEM, Gyeonggi-Do, Korea, Horiba STEC KOREA, Ltd.), the peaks of the organic matter were detected. The EEM measured the excitation wavelength up to 240–550 nm, at intervals of 2 nm, and the emission wavelength from 246.28–828.25. Multivariate data analysis techniques have been extensively used to quantitatively compare the F-EEM of samples. Multidimensional fluorescence analysis is an advanced statistical and computational technology, and parallel factor analysis (PARAFAC) is currently the most advanced technology and is commonly used. PARAFAC breaks down the sample’s EEM into multiple independent fluorescent elements.

## 3. Results and Discussion

### 3.1. Water Quality of Raw and Pretreated Water

Table 4 summarizes the water quality parameters for the feed solutions under different pretreatment methods. The raw feed water (WW) results in a turbidity of 1.3 NTU; a TOC of 19 mg/L; a UV_254_ of 0.514 cm^−1^; and a SUVA of 2.705 L/mg-m. Although the turbidity is not high, the concentration of organic matter is relatively high. This is because the organic matters in the BWRO feed were concentrated by 4~5 times in the wastewater reclamation plant. As expected, the use of CF was not effective at improving the water quality. On the other hand, the MF and UF resulted in better water qualities than the CF. While the turbidity removal by the CF was less than 4%, those by the MF and UF were 38 and 54%, respectively.

The TOC removal by the UF (32%) was higher than that by the MF (6%). The removal of UV_254_ by the UF (44%) was also significantly higher than that by the MF (3.3%). Interestingly, the removal of turbidity by the CF was insignificant, which implies that the turbidity was not caused by large suspended solids, but by small colloids and macromolecules. It is also worth noting that the removal of organic matters by the UF is not insignificant. This suggests that there were both macromolecules and small-molecular-weight organic matter in the raw feed solution.

The application of the NF was more effective to remove organic matters than the MF and UF. The removals of TOC and UV_254_ by the NF were 91 and 97%. Considering that UV_254_ is related to the concentration of hydrophobic organic matters, the results indicate that the NF is especially effective at removing hydrophobic compounds. As a result, the SUVA for the NF-treated water showed a low SUVA value (1.01 L/mg-m). The turbidity of the NF-treated water is also low (0.15 NTU), compared with the other cases. The removal of TDS by the NF was 75%. Table 5 shows the ion concentrations of the raw feed water (WW) and the NF-pretreated water. The rejections of Cl^−^ and Na^+^ by the NF are 27 and 46%. On the other hand, the rejections of divalent ions range between 87 and 99%. It is evident from the results that the high TDS removal by the NF is attributed to its high rejection of divalent ions.

The removal efficiencies of TOC by the GAC and AFM pretreatments were relatively high (>79%), as presented in Table 4. Moreover, the turbidities of these pretreated water samples were low (<0.4 NTU). This is attributed to the fact that both pretreatment methods include adsorption and filtration mechanisms for pollutant removal. Accordingly, the GAC/AFM pretreatments appear to be more advantageous than the MF/UF pretreatments because of their higher removal efficiencies of organic matters.

### 3.2. OCV and Power Density

Figure 4 shows the OCV values of the RED system for the feed solutions under different pretreatment methods. The OCV values range from 0.92 V (no pretreatment) to 1.46 V (NF). Except for the NF, the OCV values were similar, regardless of the pretreatment types. As can be seen in Table 4, the NF is the only pretreatment to reduce the TDS of the pretreated water. This suggests that only the TDS (or electric conductivity) is an important factor that affects the OCV.

The effect of the TDS removal on the OCV can also be confirmed by comparing the model calculation with the experimental OCV data. The results are summarized in Table 6. According to the model calculation, the calculated OCV values are 1.694 for the NF pretreatment, and 1.036 for all the other pretreatments. The model was applied to interpret the effect of the water quality on the OCV. All the other operational parameters, including the flow rate and the number of cell pairs, were set to be constant. Among the various water quality parameters, the TDS was considered to be a representative parameter in the model. Even with this simplification, the model results reasonably match the experimental data. This confirms that the OCV is not significantly influenced by the water quality parameters, such as the turbidity, the TOC, and the UV_254_. Only the TDS affected the OCV value in this case.

Since the OCV is the stack voltage under zero-current conditions, it cannot fully reflect the conditions of the actual operation of the SGP generation by RED. Accordingly, the power density was also measured for the raw feed water and the pretreated water samples. The results are shown in Table 7. Similar to the OCV, the initial power density was the highest for the NF-pretreated water (NF). All the other water samples result in similar values of the initial power density. The model calculations of the power density match the experimental results well, with the maximum error of 4.8%. Again, these results confirm that the initial power density does not depend on the turbidity, the TOC, or the UV_254_. Figure 5 shows the I–V curve according to the RED operation conditions. All of the other conditions, except for the NF-treated water, have similar values.

Figure 6 depicts the dependence of the power density on the time for the RED operations with different pretreatments. The initial and final power density values of the raw feed water (WW) are 0.790 and 0.739 W/m^2^, respectively. During the 24 h operation of the RED, the rate of the power density change is calculated to −1.89 × 10^−3^ W/m^2^-h. The water samples treated by the CF, MF, and UF result in the rates of the power density change of −1.89 × 10^−3^ W/m^2^-h, −2.39 × 10^−3^ W/m^2^-h, and −2.39 × 10^−3^ W/m^2^-h, respectively, which are similar to those of the raw feed water (WW). On the other hand, the water samples treated by the NF, GAC, and AFM exhibit the rates of the power density change less than −0.443 × 10^−3^ W/m^2^-h. This implies that these pretreatments (NF, GAC, AFM) are slightly better to control the reduction in the power density than the CF, MF, and UF. Considering that the difference between the two types of pretreatments is the capability of the organic removal (TOC and UV_254_), it is recommended to select the pretreatment that can reduce the organic matters in the feed water for the RED.

### 3.3. Stack Pressure

As mentioned earlier, the differences in the power density for the water samples treated by different pretreatments were not substantial. However, the differences in the stack pressure were significant, as presented in Figure 7. The stack pressure for the raw feed water (WW) was 0.5 bar at the beginning of the RED operation. However, it increased almost linearly with time, and reached up to 3.35 bar after 24 h, which corresponds to 6.7 times the initial value. The increase in the stack pressure is attributed to the blockage of the feed channel of the ion exchange membranes in the RED system. With the increased stack pressure, the energy consumption of the RED system increases, and the membranes may be deformed or damaged. Accordingly, it should be properly managed by the pretreatment of the feed solution.

The final stack pressures for the water samples treated by the MF and UF were 2.01 and 1.76 bar, respectively. On the other hand, the CF was relatively ineffective at reducing the final stack pressure. This suggests that the removal of colloids and macromolecules is effective at reducing the stack pressure to a certain degree. Higher effects of the pretreatment on the stack pressure were found in the water samples treated by NF, GAC, and AFM, which exhibited the final stack pressures of 0.95, 1.27, and 1.46 bar, respectively. This implies that, not only colloids, but also the organic matters, should be removed by the pretreatment in order to effectively reduce the stack pressure. Table 8 presents the rates of the stack pressure increase (*r_P_*), which were calculated from the linear regressions of the stack pressure curves in Figure 7. As expected, *r_p_* was the highest (0.119 bar/h) for the WW, and the lowest (0.0188 bar/h) for the NF. It appears that *r_p_* can be used as a quantitative index to compare pretreatments in terms of their effectiveness to reduce the stack pressure.

### 3.4. Visual Observations of IEX Membranes

Figure 8 presents the photographs of the anion exchange membranes (AEMs) before and after the RED experiments. The pristine membrane (Figure 8a) does not show any color on its surface. After the RED operation using the wastewater with no pretreatment (WW), the color of the membranes was changed to brown because of the deposition of contaminants, which seem to be mainly organic matters (Figure 8b).

Depending on the types of the pretreatments, the changes in the membrane color were different. For instance, the color change of the membrane by the NF-pretreated water was the smallest (Figure 8f), which implies that the deposition of the organic matter was the minimum. As reported in Figure 7, the NF-pretreated water also results in the smallest increase in the stack pressure. The color changes by the GAC- and AFM-pretreated water samples were also relatively small (Figure 8g,h), which matches their small stack pressure increases, shown in Figure 7. Accordingly, it can be concluded that the deposition of these organic matters is the main reason for the increase in the stack pressure during the RED operation.

### 3.5. EEM and PARAFAC Analysis

To investigate the characteristics of the organic matter in the wastewater and the pretreated water samples, the EEM analysis was carried out, as illustrated in Figure 9. The raw feed water (WW) showed EEM peak patterns with the excitation wavelength ranging from 220 to 350 nm, and the emission wavelength ranging from 330 to 480 nm. After the application of the CF, MF, and UF, the intensities of the EEM peaks remain similar. The fluorescent excitation–emission matrix (EEM) analysis was performed to characterize the organic matter in the raw water and pretreated water. The peaks of the organic matter were detected. Table 9 shows the four major groups of organic matter that can be identified by their excitation and emission wavelengths. This is attributed to the fact that these pretreatments cannot sufficiently remove the organic matter in the wastewater, as is shown in Table 4. On the other hand, the NF, GAC, and AFM result in a significant reduction in the EEM peak intensities. The EEM peaks at the EMs of 400 nm to 500 nm, and the Ex of 275 nm to 400 nm, are not shown in the NF. The intensities of these peaks are much lower in the GAC and AFM than in the WW, CF, MF, and UF. This is because the removal efficiencies of the organic matter by the NF, GAC, and AFM are substantial. Accordingly, the EEM peaks were effectively reduced in these cases. 

Although the EEM results in Figure 9 can be used to qualitatively compare the concentrations of organics matters in the water samples, they cannot be directly used for quantitative analysis. Accordingly, a PARAFAC analysis was carried out for the water samples to obtain more quantitative information from the EEM results. As presented in Figure 10, three main fluorescence peaks were identified by the PARAFAC analysis. These components are: terrestrial humic-like substance (C1); microbial humic-like substance (C2); and protein-like substance (C3) (Table 9). 

The components of Component 1 (C1) are humic-like fluorophores that are terrestrially or anthropogenically generated, compared with previous studies. Component 2 (C2) is microbial humic-like fluorescence, and humic and fulvic substances, and Component 3 (C3) is described as tryptophan-like fluorophores. To evaluate the quantitative change in the F-EEM of the BWRO brine, a PARAFAC analysis was applied to all of the samples in order to analyze the loading of the major components [47,48]. After the model validation of the three components, the F_max_ (maximum fluorescence intensities) for each process is calculated. A significant correlation between the F-EEM data and the maximum fluorescence modeled by PARAFAC was also observed in a previous study [49].

Table 10 compares the PARAFAC scores for the raw feed water and the pretreated water. The highest score in the raw feed water (WW) is C1 (7.0380), followed by C2 (4.6656), and C3 (4.3001). This suggests that the terrestrial humic-like substances (C1) are the main components, which is supported by a high SUVA value (2.71 L/mg-m) of the WW (Table 4). In the CF, MF, and UF cases, the scores were not significantly changed. Only the UF could slightly reduce the scores on C1 and C3. These results match the low TOC/UV_254_ rejections by these pretreatments, which are reported in Table 4. On the contrary, the scores substantially decreased in the NF, GAC, and AFM cases. The NF was found to be effective at reducing all the scores. The GAC and AFM were less effective at reducing the scores on C1 and C3, but more efficient at reducing the scores on C2. These findings suggest that the NF is more effective at removing the terrestrial humic-like and protein-like substances in the wastewater than the GAC and AFM. On the other hand, the NF is less efficient at removing microbial humic-like substances than the GAC and AFM.

This finding may be interpreted as follows: Since the fraction of fulvic acid in the microbial humic-like substances is higher than that in the terrestrial humic-like substances [50], the microbial humic-like substances may have lower rejections by the NF membrane. On the other hand, the protein-like substances may have charges, which result in high rejections by the NF membrane [51]. Of course, an in-depth analysis of the chemical structures of these organic matters and their effect on NF rejection will be required in order to support this interpretation.

### 3.6. Correlations between Water Quality Parameters and Stack Pressure

As discussed earlier, the increase in the stack pressure is a critical problem in the RED system because it leads to an increase in the energy loss, as well as the possibility of IEM damage. Accordingly, it is necessary to predict the stack pressure by using water quality parameters. Figure 11 shows the correlation between the water quality parameters and the rate of the stack pressure increase (*r_P_*). There is a linear relationship between the turbidity and *r_P_* (*R^2^* = 0.975) in Figure 11a. As pointed out before, the turbidity of the water samples includes not only colloids, but also dissolved organics. Accordingly, it seems that the increase in the stack pressure results from the clogging of the membrane channel by the colloids and organic matter. On the other hand, the correlation between the TOC and *r_P_* is not strong (*R^2^* = 0.77), as is shown in Figure 11b. The deviation between the regression curve and the measured *r_p_* is more significant when the *r_p_* values are higher. This is because the TOC cannot represent the effect of the colloids that affects the stack pressure. The correlation between the UV_254_ and *r_P_* is also weak (*R^2^* = 0.78), as is illustrated in Figure 11c. Again, the UV_254_ is less appropriate than the turbidity for predicting *r_p_* because it is not a direct measure for colloidal substances. 

The scores that were calculated from the PARAFAC analysis were used to obtain the correlations with *r_p_*. As is illustrated in Figure 11d,f, the C1 and C3 result in weak correlations with *r_p_*. Their R^2^ values are 0.72 and 0.76, respectively, which are similar to those of the TOC and the UV_254_. Interestingly, the correlation between the C2 and *r_p_* was poor (*R^2^* = 0.43), as is shown in Figure 11e. These results suggest that the increase in the stack pressure is affected by the terrestrial humic-like organic matters (C1) and protein-like organic matters (C3) but is independent of microbial humic-like organic matters (C2). The differences in the physicochemical properties among these organic matters may result in different stack pressures. Nevertheless, further works will be required in order to further understand the correlation between the properties of the organic matter and their affinity to the IEMs and spacers in the RED stack.

## 4. Conclusions

In this study, the effect of pretreatment on the OCV, the power density, and the stack pressure was investigated in a bench-scale RED system, which used RO brine from a wastewater reclamation plant as the low-salinity (LS) solution, and synthetic SWRO brine as the high-salinity (HS) solution. Six pretreatment methods were applied, including the CF, MF, UF, NF, GAC, and AFM. The following conclusions were drawn:The RO brine without pretreatment had a relatively high TOC (19 mg/L) and UV254 (0.514 cm^−1^), while the CF, MF, and UF could not reduce the organic matters; the NF, GAC, and AFM showed the TOC removal ranging from 79 to 91%, and the UV_254_ removal ranging from 83% to 97%;The OCV value for the NF-pretreated water was 1.46 V, and the OCV values for all the other cases were in the range between 0.92 V and 0.97 V. The OCV is not significantly influenced by the turbidity, the TOC, and the UV_254_, but it is by the TDS;Similar to the OCV, the power density was higher for the NF-pretreated water (1.15 W/m^2^) than for the other cases (0.79 ~ 0.8 W/m^2^). The reduction in the power density with time was not significant (<−2.39 × 10^−3^ W/m^2^-h, less than 15% per 24 h). The NF, GAC, and AFM were slightly better at controlling the reduction in the power density than the CF, MF, and UF;The experimental results on the OCV and power density for the water samples were matched well with the model calculations. The errors of the OCV calculations range between 6.3 and 13.8%. Those of the power density calculations range from 3.6% to 4.8%;Although the turbidity of the untreated feed (RO brine without pretreatment) was not high (1.3 NTU), the stack pressure increased from 0.5 to 3.35 bar within 24 h. The final stack pressures for the water samples treated by the CF, MF, and UF were higher than those treated by the NF, GAC, and AFM;The PARAFAC analysis was carried out for the water samples with different pretreatments. Three main fluorescence peaks were identified by the PARAFAC analysis, including terrestrial humic-like substance (C1), microbial humic-like substance (C2), and protein-like substance (C3). In the CF, MF, and UF cases, the scores were not significantly changed. On the contrary, the scores substantially decreased in the NF, GAC, and AFM cases;The rates of the stack pressure increase were correlated with the water quality parameters and the PARAFAC scores. The correlation between the turbidity and the increase in the stack pressure was the strongest. There were also reasonable relationships between the rates of the stack pressure increase and C1/C3. On the other hand, the rates of the stack pressure increase were not successfully correlated with C2. These imply that the increase in the stack pressure is closely related to the amounts of terrestrial humic-like substances and protein-like substances;Although the NF exhibited the highest pretreatment efficiency, it uses a substantial amount of energy, which leads to a reduction in the net energy production by RED. Accordingly, the GAC and AFM are recommended as the optimum RED pretreatment methods because of their effectiveness at removing organic matter.

## Figures and Tables

**Figure 1 membranes-12-00372-f001:**
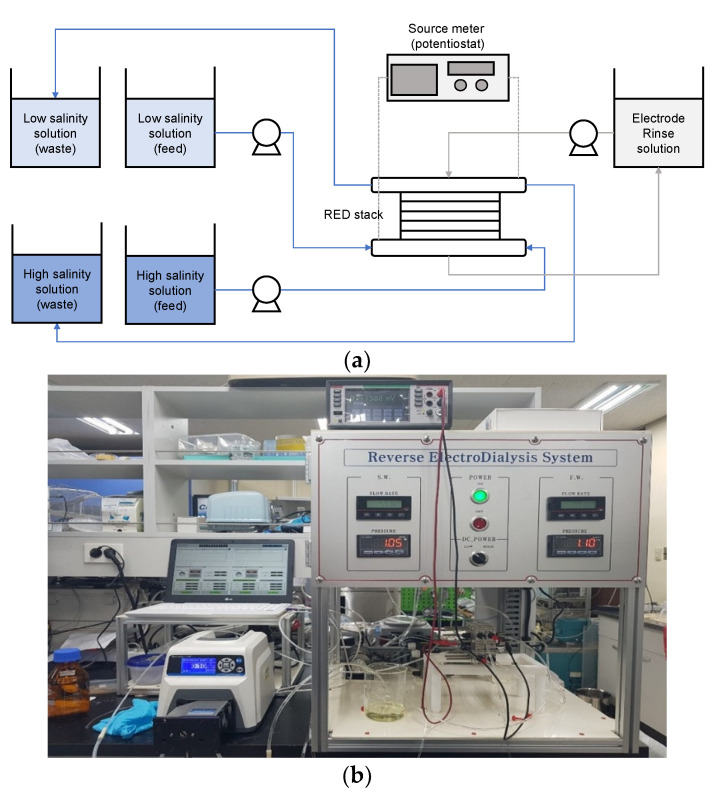
(**a**) Schematic diagram of RED experimental equipment; and (**b**) photography of the experimental setup.

**Figure 2 membranes-12-00372-f002:**
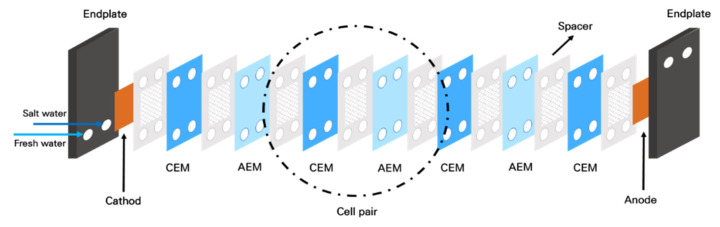
Schematic diagram of the RED stack.

**Figure 3 membranes-12-00372-f003:**
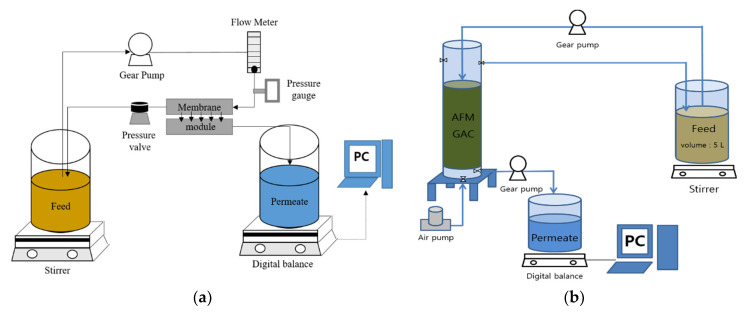
Schematic diagram of bench-scale pretreatment device: (**a**) CF/MF/UF/NF; and (**b**) AFM/GAC.

**Figure 4 membranes-12-00372-f004:**
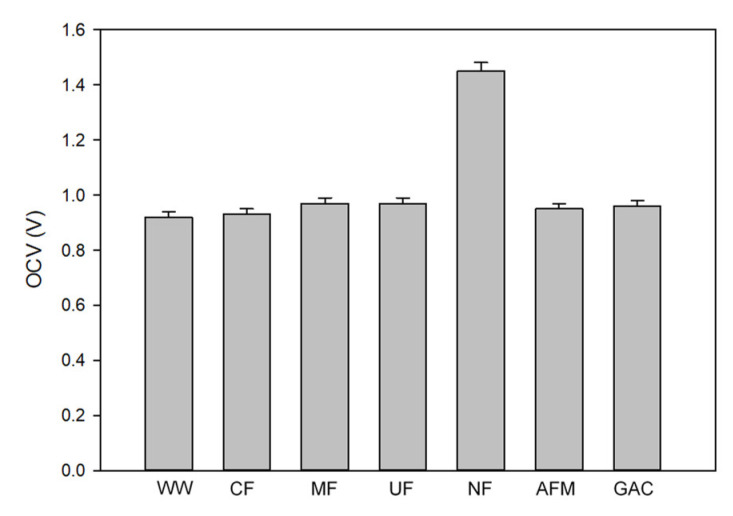
OCV values of RED system with different pretreatments (WW: raw feed solution; CF: cartridge filtration; MF: microfiltration; UF: ultrafiltration; NF: nanofiltration; AFM: activated filter media; GAC: granular activated carbon).

**Figure 5 membranes-12-00372-f005:**
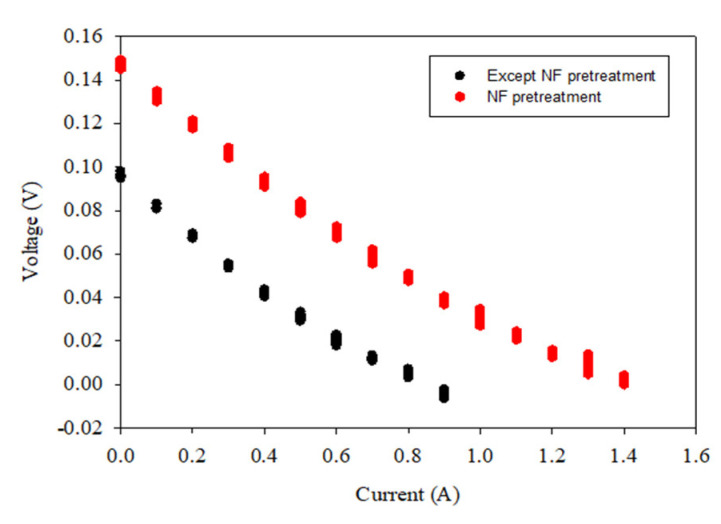
V–I curve according to the RED operation (the operation conditions are except NF pretreatment and NF pretreatment each).

**Figure 6 membranes-12-00372-f006:**
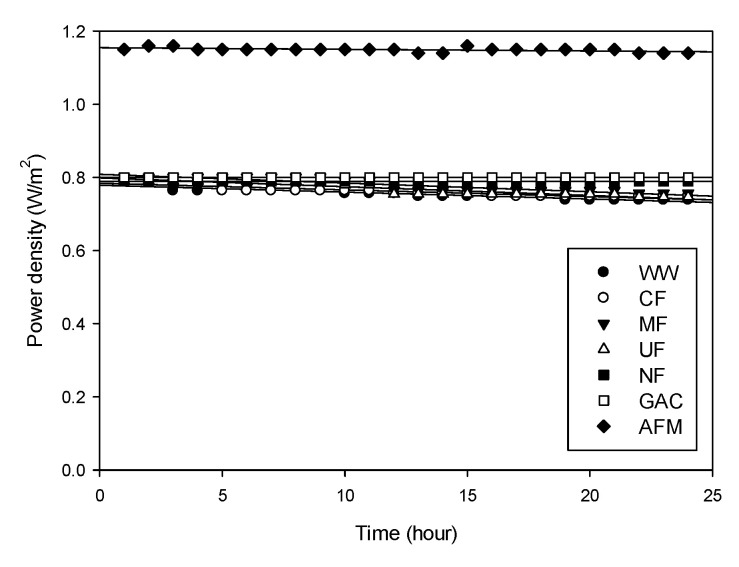
Changes in power density with time for feed solutions with different pretreatments.

**Figure 7 membranes-12-00372-f007:**
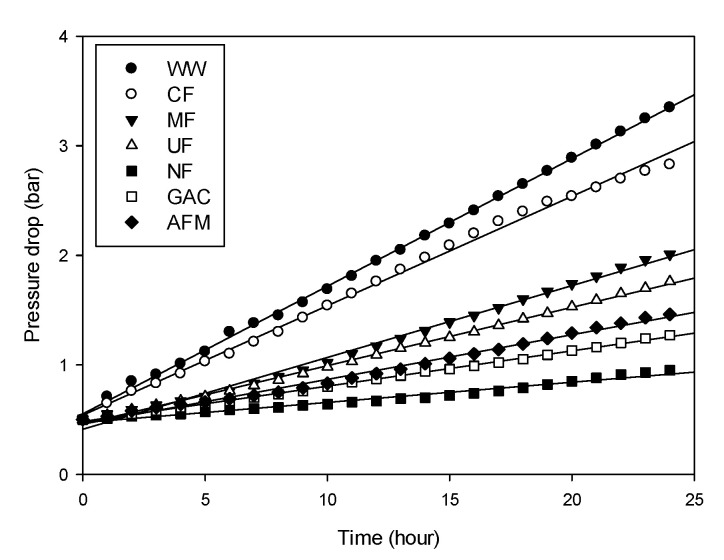
Changes in stack pressure with time for feed solutions with different pretreatments.

**Figure 8 membranes-12-00372-f008:**
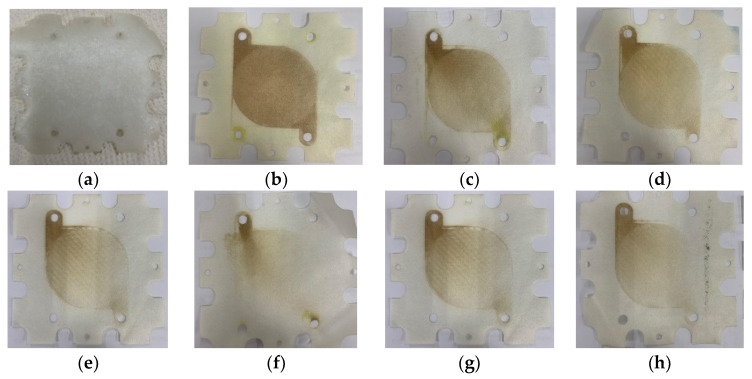
Visual observation of AEMs: (**a**) pristine AEM; (**b**) AEM using WWBR; (**c**) AEM using WWBR pretreated by CF; (**d**) AEM using WWBR pretreated by MF; (**e**) AEM using WWBR pretreated by UF; (**f**) AEM using WWBR pretreated by NF; (**g**) AEM after the use of AFM; and (**h**) AEM after the use of GAC.

**Figure 9 membranes-12-00372-f009:**
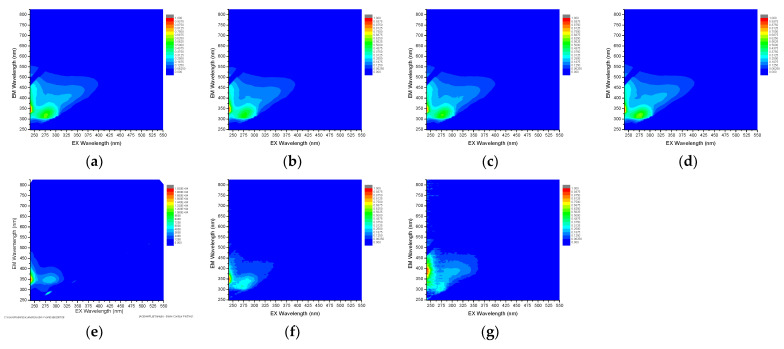
EEM data with different pretreatments: (**a**) WW: wastewater brine; (**b**) CF: cartridge filtration; (**c**) MF: microfiltration; (**d**) UF: ultrafiltration; (**e**) NF: nanofiltration; (**f**) AFM: activated filter media; and (**g**) GAC: granular activated carbon.

**Figure 10 membranes-12-00372-f010:**

Contour plots of three components identified by the PARAFAC model: (**a**) C1; (**b**) C2; and (**c**) C3.

**Figure 11 membranes-12-00372-f011:**
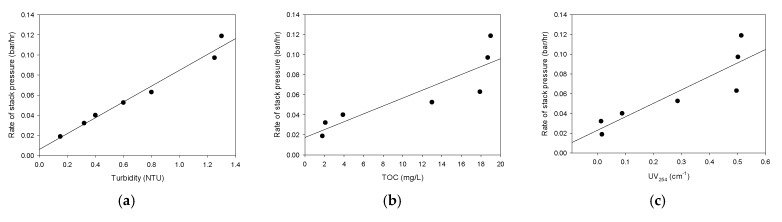
Correlations between water quality parameters and stack pressure: (**a**) turbidity; (**b**) TOC; (**c**) UV254; (**d**) PARAFAC Score 1; (**e**) PARAFAC Score 2; (**f**) PARAFAC Score 3.

**Table 1 membranes-12-00372-t001:** Operating conditions for RED experiments.

Conditions	Values
Cell pairs (stack)	10
Area of one membrane (m^2^)	0.0019
Q_HC_ (mL/min)	15
Q_LC_ (mL/min)	15
C_HC_/C_LC_ (M)	0.6 M/0.1 M
Temperature (K)	293

**Table 2 membranes-12-00372-t002:** Experimental conditions for membrane fabrication.

Conditions		Specifications
Manufacture	CEM	Fujifilm (Type-1, Manufacturing Europe, The Netherlands)
	AEM	
Thickness (μm)	CEM	125
	AEM	124
Area resistance (Ω·cm^2^)	CEM	1.87 ± 0.01
	AEM	1.08 ± 0.02
Transport number (-)	CEM	0.952
	AEM	0.963

**Table 3 membranes-12-00372-t003:** Summary of pretreatment methods.

	CF	MF	UF	NF	GAC	AFM
Manufacturer	Millipore	Millipore	A/G technology	Dow	Sunghong-Lab	Dryden Aqua
Model	TMTP14250	GVHP 14250	UFP10	NF 70	Granular activated carbon	Activated filter media
Pore size (μm)	5	0.22	100 kDa (MWCO)	-	-	-
Media size (mm)	-	-	-	-	0.2~5	0.4~1
Feed flow rate (L/min)	0.5	0.5	0.5	0.5	0.5	0.5
Applied pressure (bar)	0.1	0.5	1	2.5	0.08	0.08

**Table 4 membranes-12-00372-t004:** Water quality of raw feed water (WW) and pretreated water.

	Electric Conductivity (μS/cm)	Turbidity (NTU)	TOC (mg/L)	UV_254_ (cm^−1^)	SUVA (L/mg-m)
WW (no pretreatment)	5850	1.30	19.0	0.514	2.71
CF	5850	1.25	18.7	0.502	2.68
MF	5850	0.80	17.9	0.497	2.94
UF	5850	0.60	13.0	0.287	2.20
NF	1456	0.15	1.8	0.017	1.01
GAC	5850	0.32	2.1	0.014	1.27
AFM	5850	0.40	3.9	0.089	2.28

**Table 5 membranes-12-00372-t005:** Ion concentrations of raw feed water (WW) and NF-pretreated water.

	Chloride (mg/L)	Sulfate (mg/L)	Sodium (mg/L)	Calcium (mg/L)	Magnesium (mg/L)	Potassium (mg/L)	Silica (mg/L)
WW	111.48	565	798	1191	232	164	33.1
NF	81.1	53.5	431	10.8	30.3	13.6	8.7

**Table 6 membranes-12-00372-t006:** Experimental and calculated OCVs in RED using raw feed water (WW) and pretreated water.

Water Type	Experimental OCV (V)	Calculated OCV (V)	Error (%)
WW (no pretreatment)	0.92	1.036	11.19
CF	0.93	1.036	10.23
MF	0.97	1.036	6.3
UF	0.97	1.036	6.3
NF	1.46	1.694	13.8
AFM	0.95	1.036	8.3
GAC	0.96	1.036	7.3

**Table 7 membranes-12-00372-t007:** Power density in RED using raw feed water (WW) and pretreated water.

Water Type	Experimental Power Density (W/m^2^)	Calculated Power Density (W/m^2^)	Error (%)
WW (no pretreatment)	0.790	0.83	4.8
CF	0.790	0.83	3.6
MF	0.800	0.83	3.6
UF	0.800	0.83	3.6
NF	1.15	1.2	4.1
AFM	0.790	0.83	3.6
GAC	0.800	0.83	3.6

**Table 8 membranes-12-00372-t008:** Stack pressure in RED using raw feed water (WW) and pretreated water.

Water Type	Initial Stack Pressure (bar)	Final Stack Pressure (bar)	Rate of Stack Pressure Increase (bar/h)
WW (no pretreatment)	0.5	3.35	0.11875
CF	0.5	2.83	0.097083
MF	0.5	2.01	0.062917
UF	0.5	1.76	0.0525
NF	0.5	0.95	0.01875
AFM	0.5	1.27	0.032083
GAC	0.5	1.46	0.04

**Table 9 membranes-12-00372-t009:** Fluorescing components determined by PARAFAC model [47,48,49].

Components	Ex/Em	Description
Component 1 (C1)	250(350)/450	Terrestrial humic-like fluorescence
Component 2 (C2)	250(325)/400	Microbial humic-like fluorescence
Component 3 (C3)	275/306	Tryptophan-like substances (protein-like)

**Table 10 membranes-12-00372-t010:** Scores on three components in PARAFAC model for raw feed water (WW) and pretreated water.

Water Type	Scores on Component 1	Scores on Component 2	Scores on Component 3
WW (no pretreatment)	7.0380	4.6656	4.3001
CF	6.5034	4.2204	4.1537
MF	6.8197	4.5548	4.0824
UF	6.2786	4.9669	3.7588
NF	3.2413	2.7326	1.9000
AFM	4.0954	1.3516	2.5199
GAC	4.3546	1.5523	2.7844

## Data Availability

Not applicable.

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
