# Peer review of "Comparison of Pretreatment Methods for Salinity Gradient Power Generation Using Reverse Electrodialysis (RED) Systems"

_membranes, 2022, doi:10.3390/membranes12040372_

Round 1

Reviewer 1 Report

The purpose of the article is to identify suitable pre-treatment methods to remove organic/inorganic foulants from the wastewater plant RO retentate used as the low salinity feed for the RED for salinity gradient power. The authors have used commercial membranes in an custom built RED stack to measure the OCV and the power density using synthetic seawater brine and the wastewater RO retentate as the feed solutions of different salinities. Different pre-treatment methods, categorized as membrane based and media based filtrations, have been tested to identify the optimal pre-treatment techniques to reduce fouling of ion exchange membranes, that leads to increase in stack pressure and eventual loss of stack performance. 

Overall, the work leads to meaningful observations that are useful in actual RED process and the drawn conclusions correlate well with the experimental data. However, there are some gaps in describing experimental methods, explaining the problem statement in more detail and also in some experimental data interpretation. I recommend that the manuscript be accepted for publication once the authors have addressed these issues, listed below. 

  1. Line 48, I think the authors meant PRO but mention RED. Please change "RED" to "PRO"
  2. Introduction section: The problem statement must be explained in more detail. the typical problem with using BWRO brine (from wastewater treatment plants) is the contaminants that foul the membranes in both PRO and RED. The purpose of the article needs to be clearly elucidated, i.e. Why is organic fouling a significant issue not just in RED but also in PRO, which leads to performance loss and why it is critical to identify the most optimal pre-treatment method to reduce organic fouling.  There are several references  available on this topic. e.g. that deal with pre-treatment methods. Some examples include.  
    Water Research 103 (2016) 264-275
    Water Research 88 (2016) 144-155
    Journal of Membrane Science 572 (2019) 658–667.
  3. There are several instances where "Cells pair" was mentioned instead of "Cell Pairs". please revise. 
  4. The quality of Figure 1(a), schematic diagram of the RED stack is not good. Authors may consider presenting a simple process flow diagram instead but the components of the stack must be clear. Also, ERS is not defined anywhere. Please add the description "Electrode Rinse Solution" the first time the term is used. 
  5. Line 189, please capitalize the word "red" to "RED".
  6. Line 222, please change "power" to "power density".  also 

    The equation connecting the OCV to the power density is missing. E = OCV - R(x).I
    What is the internal stack resistance used for the calculations?

  7. Table 4. Total raw water analysis is required to understand the fouling/scaling potential. Is there any silica present which is also a known inorganic foulant? From the NF removal data, it appears that majority of the pollutants are divalent ions, which is unusual because the BWRO pretreatment must have taken care of scalants. It is important to look at the ionic constituents to explain this high TDS removal because NF can remove most divalent ions and some monovalent ions. 
  8. Line 266. The model only considers the effect of TDS on the OCV. How accurate is this assumption? There are other operational parameters that effect the OCV and the Pmax. Like the flow rate or number of cell pairs, etc. Please explain. 
  9. Line 348. There is no mention of the experimental procedure, equipment details  or sampling method for the EEM spectra. Please include this in the materials and methods section. 
  10. Line 354. Can authors explain how the data can be interpreted quantitatively? which peak intensities are they referring to? is it possible to represent the wavelength vs intensity data in a graphical form or tabular form to interpret these images more quantitatively? Only (e) and (f) appear to be different from other images. it is not clear how it is deduced that GAC treatment results in significant reduction in intensities. 
  11. Table 8. Need reference(s) for the wavelengths used for C1, C2 and C3., unless they were measured in this study.
  12. Line 380. need references here. Also, more explanation is needed for this observation as well as what the author's understanding of NF membrane systems is. 
  13. Line 412-413. Deposition tendency or fouling propensity are not dependent on molecular weight alone. They also depend on the chemical functionalities, surface morphology, and other factors. This sentence should be revised and references should be added if any such similar behavior was observed before.

Author Response

We are grateful to reviewer for the valuable comments and useful suggestions that have helped us to improve our paper considerably. As indicated in the following responses, we have incorporated all these comments into the revised version of our paper

Reviewer 2 Report

This article study that mainly several pretreatments applied to the RED process, including cartridge filter (CF), microfiltration (MF), ultrafiltration (UF), nanofiltration (NF), activated filter media (AFM), and granular activated carbon (GAC).

  1. Please check and correct thoroughly the subscript of the abbreviation in the text, such as Nm in Eq. (1), etc.
  2. Where does the raw feed water come from? What are the specific ingredients of the colloidal and organic matters?
  3. Please provide the IV curve of the RED operation in the "3.2. OCV and power density" section
  4. Please complete the list of symbols.
  5. References lack work from 2020. Multiple references are from the same author, please pay attention to the relevance and comprehensiveness of the references.

Author Response

(The authors gave the same response as above.)
